# Efficient alkane oxidation under combustion engine and atmospheric conditions

Zhandong Wang [1,2,3,9 ✉], Mikael Ehn [4,9], Matti P. Rissanen [4,5], Olga Garmash [4], Lauriane Quéléver[4], Lili Xing[6], Manuel Monge-Palacios [3], Pekka Rantala [4], Neil M. Donahue [7], Torsten Berndt [8] & S. Mani Sarathy [3 ✉]

Oxidation chemistry controls both combustion processes and the atmospheric transformation of volatile emissions. In combustion engines, radical species undergo isomerization reactions that allow fast addition of $O_2$. This chain reaction, termed autoxidation, is enabled by high engine temperatures, but has recently been also identified as an important source for highly oxygenated species in the atmosphere, forming organic aerosol. Conventional knowledge suggests that atmospheric autoxidation requires suitable structural features, like double bonds or oxygen-containing moieties, in the precursors. With neither of these functionalities, alkanes, the primary fuel type in combustion engines and an important class of urban trace gases, are thought to have minor susceptibility to extensive autoxidation. Here, utilizing state-of-the-art mass spectrometry, measuring both radicals and oxidation products, we show that alkanes undergo autoxidation much more efficiently than previously thought, both under atmospheric and combustion conditions. Even at high concentrations of $NO_X$, which typically rapidly terminates autoxidation in urban areas, the studied $C_6$–$C_{10}$ alkanes produce considerable amounts of highly oxygenated products that can contribute to urban organic aerosol. The results of this inter-disciplinary effort provide crucial information on oxidation processes in both combustion engines and the atmosphere, with direct implications for engine efficiency and urban air quality.

---

[1] National Synchrotron Radiation Laboratory, University of Science and Technology of China, Hefei, Anhui 230029, P. R. China. [2] State Key Laboratory of Fire Science, University of Science and Technology of China, Hefei, Anhui 230026, PR China. [3] King Abdullah University of Science and Technology (KAUST), Clean Combustion Research Center (CCRC), Thuwal 23955-6900, Saudi Arabia. [4] Institute for Atmospheric and Earth System Research (INAR), University of Helsinki, Helsinki 00014, Finland. [5] Aerosol Physics Laboratory, Physics Unit, Faculty of Engineering and Natural Sciences, Tampere University, 33720 Tampere, Finland. [6] Energy and Power Engineering Institute, Henan University of Science and Technology, Luoyang, Henan 471003, China. [7] Center for Atmospheric Particle Studies, and Department of Chemistry, Department of Chemical Engineering, Department of Engineering and Public Policy, Carnegie Mellon University, Pittsburgh, PA 15213, USA. [8] Leibniz Institute for Tropospheric Research (TROPOS), Atmospheric Chemistry Dept. (ACD), 04318 Leipzig, Germany. [9] These authors contributed equally: Zhandong Wang, Mikael Ehn. ✉email: zhdwang@ustc.edu.cn; mani.sarathy@kaust.edu.sa

Autoxidation plays an important role in a wide range of chemical systems[1–6], contributing to the spoilage of food and wine, ignition in internal combustion engines, and formation of atmospheric organic aerosol (OA) from volatile emissions. Autoxidation chemistry is outlined in more detail in Supplementary Note 1 and Supplementary Scheme 1. The key feature is that oxidation is driven directly by molecular oxygen ($O_2$) after some initiation step; while this phenomenology has been understood for some time, the mechanistic details have not. Autoxidation is initiated by formation of peroxy radicals (ROO•) and propagated via H-atom transfers ("H-shift" isomerization), forming carbon-centered radicals, ROO• → •QOOH[7]. Subsequently, $O_2$ adds to form a new peroxy radical, •QOOH + $O_2$ → •OOQOOH. Under suitable conditions, this process (H-shift and $O_2$ addition) may repeat to form progressively more oxygenated ROO•. The propensity of a system to undergo such "multi-step autoxidation" governs fuel ignition timing in engines and largely determines the potential of atmospheric volatile organic compounds (VOC) to form low-volatility condensable vapors, and consequently OA. The extent to which multi-step autoxidation occurs before radical termination is determined by the reactant's molecular structure, the reaction temperature and pressure, and the presence of bimolecular reaction partners. Determining the fate of ROO• (henceforth $RO_2$) is thus critical to quantifying key properties of reaction products in a diverse set of systems, ultimately affecting both human health[8] and climate[9].

In this work, we explore the autoxidation of alkanes, an important class of molecules found in fuels and also ubiquitous in urban atmospheric environments[10]. Our research was motivated by three recent scientific advances. First, developments in chemical ionization mass spectrometry, CIMS, have made the sensitive detection of autoxidation products possible (e.g., peroxy radicals and closed shell species, including carbonyls, nitrates, and accretion products). In particular, the Chemical Ionization Atmospheric Pressure interface Time of Flight mass spectrometer, CI-APi-TOF (see "Methods") has been especially powerful, enabling discovery of highly oxygenated organic molecules (HOM, with 6 or more O-atoms, Supplementary Note 1)[4,11], with molar yields on the order of a few percent from several biogenic (e.g., monoterpenes) and anthropogenic (e.g. aromatics) VOC[12,13]. While seemingly minor (from the perspective of carbon balance), in the atmosphere this HOM channel often drives the formation of secondary OA, SOA[4,14] and even new-particle formation[15]. No studies on HOM formation from alkanes have been reported, potentially due to assumptions that their structures would be unfavorable for autoxidation.

Second, alkane fuel autoxidation under combustion conditions was only recently shown to extend to three sequential $O_2$ addition steps[6], significantly altering the conventional understanding of fuel ignition chemistry. Elucidating whether further autoxidation occurs, to form even more complex oxygenated ignition precursors, has been nearly impossible with the sensitivities of the analytical methods typically utilized in combustion research.

Third, disparate SOA yields have been reported for various alkane types upon reaction with OH radicals. Studies consistently report rising SOA yields for alkanes with the same carbon number, in the order cyclic > linear > branched[16,17]. This structural dependence has been attributed to different reaction pathways of alkoxy (RO) radicals[17], which are common and highly reactive intermediates in atmospheric oxidation, formed from bimolecular reactions of $RO_2$ (see Supplementary Note 2 and Supplementary Scheme 2–5 for RO chemistry details). Briefly, the general rationale is that RO from branched alkanes undergo fragmentation via C–C bond scissions to form more volatile products, RO from linear alkanes (usually with 6 or more carbon atoms) undergo an H-shift isomerization, and RO from cycloalkanes undergo C–C bond scission without fragmentation. The products formed from both linear and cyclic alkanes will reform $RO_2$ radicals, though the cycloalkanes will have acquired an additional aldehyde functionality, and the linear alkanes will have acquired a hydroxyl functionality (Supplementary Note 2). The dramatic increase in SOA yields between linear and cyclic alkanes of the same carbon chain length is hard to explain simply by this difference[17]. We hypothesize that autoxidation has been overlooked in these systems; this may explain many of the aforementioned differences in observed SOA yields.

## Results and discussion

Figure 1 summarizes important initial reaction pathways in alkane oxidation, including autoxidation, both in the combustion and atmospheric domains. Gray colors show the radical reactions and intermediates reported previously[6,18], while black colors show the results that will be presented in this study. In combustion engines at T > 500 K, H-abstraction from the fuel alkane occurs by reaction with $O_2$, while in the atmosphere at T ≈ 300 K, H-abstraction occurs primarily by OH radicals. The resulting alkyl radical (R) rapidly adds $O_2$ to form the initial $RO_2$ radicals. Under combustion conditions, subsequent multi-step autoxidation occurs via H-shift and $O_2$ addition to form higher molecular weight $RO_2$; the extent of autoxidation is limited almost exclusively by competing unimolecular termination via loss of OH or $HO_2$ from the $RO_2$. Under atmospheric conditions, autoxidation is limited primarily by bimolecular reactions involving $RO_2$ radicals, mainly with NO, $HO_2$ and other $RO_2$ radicals (Supplementary Note 2). Bimolecular reactions are chain-terminating except when RO radicals are formed. Further oxygen additions are possible when the RO decays either via an H-shift or, in the case of cycloalkanes, via a C–C bond scission.

**From the combustion domain to the atmospheric domain.** We investigated the products formed from the autoxidation of linear, branched, and cyclic alkanes, as well as their common oxygenated derivatives (carbonyls and alcohols) in different systems. Table 1 summarizes all the experiments carried out in this work.

Initially, we coupled a jet-stirred reactor (JSR) working at a combustion relevant reaction temperature to a CI-APi-TOF using $NO_3^-$ as the reagent ion (see "Methods", Supplementary Note 3). The mass spectra measured during oxidation of n-decane, 2,7-dimethyloctane, n-butylcyclohexane, and 2-decanone at 545 K are shown in Supplementary Figure 2, while that of decanal at 520 K is shown in Fig. 2a. Table 2 summarizes representative products from the combustion autoxidation of these alkanes and their oxygenated derivatives, as measured previously by photoionization mass spectrometry (SVUV-PIMS)[6] and in this work by CI-APi-TOF.

Prior work has suggested a modest degree of autoxidation for decanal[6], and here we use this oxygenated species as a reference for autoxidation over the temperature sequence from 520 to 300 K. During autoxidation at 520 K, decanal (Fig. 2a) shows a prominent signal at $C_{10}H_{18}O_6$, confirming our previous observation by SVUV-PIMS[6] of three sequential $O_2$ addition steps (Supplementary Fig. 3). However, additional peaks corresponding to $C_{10}H_{18}O_8$ and $C_{10}H_{18}O_{10}$ indicate a fourth and even fifth $O_2$ addition. These products contain multiple –OOH groups, and represent previously undiscovered intermediates that, upon decomposition, result in radical chain-branching. This finding significantly enriches our understanding of autoignition under engine relevant conditions, showing that autoxidation continues to a significantly greater extent than previously thought for both alkanes and their oxygenated derivatives (Supplementary Scheme 6).

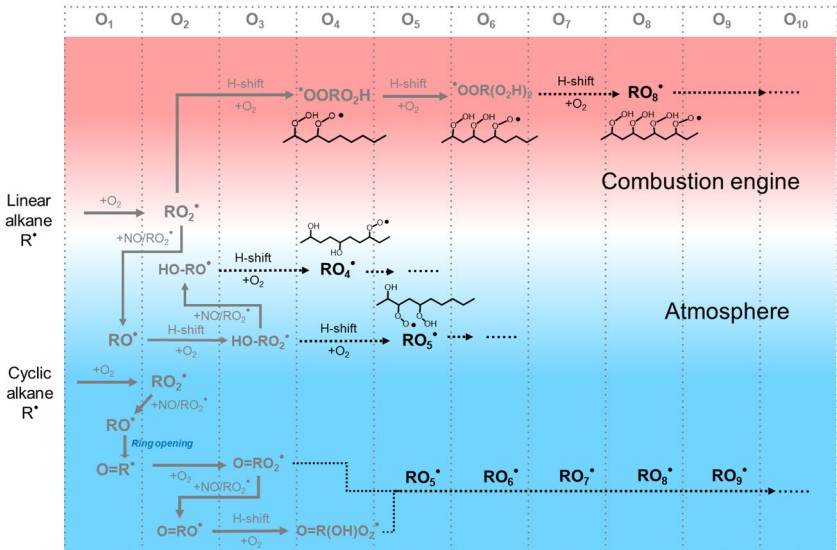

**Fig. 1 General reaction mechanisms of peroxy radicals, RO$_2$, from alkane oxidation at combustion-relevant (T > 500 K, red region) and atmospheric (T ≈ 300 K, blue region) conditions.** The figure focuses on possible radical propagation pathways, omitting termination reactions. For linear alkanes, *n*-decane is used to illustrate some hypothetical example structures. Gray colors depict previously reported radical reactions and intermediates[6,18], while the black summarize the results of this study, showing highly oxygenated compounds identified for the first time. The vertical grids separate molecules with different O-atom content. The notations "RO$_{X>2}$" refer to peroxy radicals with a total of X O-atoms.

---

**Table 1 Overview of the volatile organic compounds (VOC) used in this work.**

| Class | Name | Formula | Structure | JSR-UHEL | FR-UHEL | FR-TROPOS |
|---|---|---|---|---|---|---|
| Alkane | *n*-decane | $C_{10}H_{22}$ | | x | x | x |
| | 2,7-dimethyloctane | $C_{10}H_{22}$ | | x | x | |
| | cyclohexane | $C_6H_{12}$ | | | | x |
| Cycloalkane | methylcyclohexane | $C_7H_{14}$ | | | x | |
| | *n*-butylcyclohexane | $C_{10}H_{20}$ | | x | x | |
| | cis-decalin | $C_{10}H_{18}$ | | | x | x |
| | trans-decalin | $C_{10}H_{18}$ | | | x | x |
| Oxygenate | heptanal | $C_7H_{14}O$ | | | x | |
| | decanal | $C_{10}H_{20}O$ | | x | x | |
| | 1-decanol | $C_{10}H_{22}O$ | | | x | |
| | 2-decanone | $C_{10}H_{20}O$ | | x | | |

The last three columns refer to the three different types of experiment conducted.
Note: JSR-UHEL: jet-stirred reactor experiments at University of Helsinki; FR-UHEL: flow reactor experiments at University of Helsinki; FR-TROPOS: flow reactor experiment at Leibniz Institute for Tropospheric Research.

---

Next, we progressively lowered the JSR temperature to explore the transition from combustion to atmospheric conditions. We used the reaction of tetramethyl ethylene (TME) with O$_3$ to generate OH radicals to initiate the oxidation sequence. At T = 392 K (Fig. 2b), the monomer spectra are similar to those at 520 K, but now ROOR accretion products[19] emerge with signals at C$_{20}$H$_{38}$O$_{6,8,10,12,14}$. From T = 392 K to 334 K (Fig. 2c), all observed signals decrease, especially those of the most oxygenated

molecules. Radical species with odd hydrogen numbers—C$_{10}$H$_{19}$O$_{5,6,7}$—also appear and the spectrum now shows ROOR accretion products with molecular formulas separated by one O atom rather than by O$_2$. All of these changes suggest less efficient H-shifts in the RO$_2$ at lower temperatures, and an increasing role of at least one H-shift in RO radicals leading to the formation of highly oxidized species (Supplementary Note 4 and Supplementary Scheme 7). We note that the exact oxidation rates are

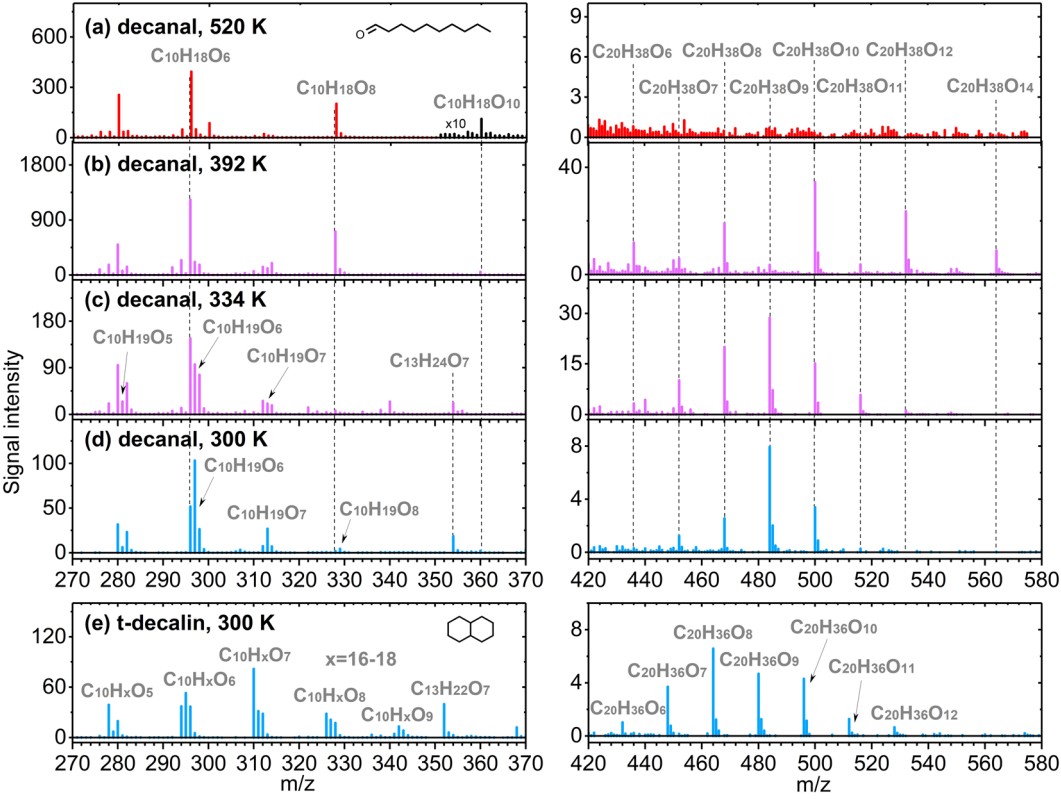

**Fig. 2 Product species distribution from the autoxidation of VOCs as measured by mass spectrometry.** The mass spectra for decanal are in (**a**–**d**) and the mass spectrum for trans-decalin is in (**e**). The left and right panels cover the mass ranges of monomer and ROOR accretion products, respectively. TME + $O_3$ reactions were used to produce OH radicals needed to initiate VOC oxidation in all experiments, except at T = 520 K, where oxidation was initiated by $O_2$. $C_{13}$ products result from accretion between TME derived $C_3$_$RO_2$ and decalin/decanal derived $C_{10}$_$RO_2$. Spectra in **a**–**c** were measured from the JSR, **d**–**e** from the Helsinki flow reactor.

**Table 2 Representative products detected during combustion autoxidation (545 K) of alkanes and their oxygenates.**

| Molecule | Structure | $O_2$ addition sequence | | | | |
|---|---|---|---|---|---|---|
| | | 1st[a] | 2nd[a] | 3rd[a, b] | 4th[b] | 5th[b] |
| n-decane | | $C_{10}H_{20}O$ | $C_{10}H_{20}O_3$ | $C_{10}H_{20}O_5$ | $C_{10}H_{20}O_7$ | – |
| 2,7-dimethyloctame | | $C_{10}H_{20}O$ | $C_{10}H_{20}O_3$ | $C_{10}H_{20}O_5$ | $C_{10}H_{20}O_7$ | – |
| n-butylcyclohexane | | $C_{10}H_{18}O$ | $C_{10}H_{18}O_3$ | $C_{10}H_{18}O_5$ | $C_{10}H_{18}O_7$ | – |
| 2-decanone | | $C_{10}H_{18}O_2$ | $C_{10}H_{18}O_4$ | $C_{10}H_{18}O_6$ | $C_{10}H_{18}O_8$ | – |
| Decanal | | $C_{10}H_{18}O_2$ | $C_{10}H_{18}O_4$ | $C_{10}H_{18}O_6$ | $C_{10}H_{18}O_8$ | $C_{10}H_{18}O_{10}$ |

The listed molecules correspond to the carbonyl (R=O) termination products of $RO_2$ radicals. For all compounds a 4th $O_2$ addition was observed, and for decanal even a 5th $O_2$ addition took place.
[a]Measured by SVUV-PIMS in Wang et al.[6]
[b]Measured by CI-APi-TOF in this work.

unknown in Fig. 2a–c, so they should not be compared quantitatively.

At atmospheric conditions (~300 K), relative humidity is <1%, we studied the oxidation of 6 alkanes and also 3 oxygenates (Table 1, Supplementary Note 5) using a flow reactor setup at the University of Helsinki ("Methods"). The 3 s residence time enables to follow the initial steps of the autoxidation reactions. From this set of experiments, despite the short reaction time and low temperature, high signals of highly oxygenated species are still observed. This is unexpected, as $RO_2$ H-shifts in alkanes are reportedly slow and should rapidly lead to radical termination[20,21]. Decanal (Fig. 2d) and the bicyclic alkane decalin

(Fig. 2e) show the highest product signals, but HOM are evident from all tested VOC (Supplementary Figs. 5, 6, 8–12), except for n-decane and 2,7-dimethyloctane (Supplementary Fig. 13). We estimated molar HOM yields (Supplementary Note 6) at T = 300 K for all HOM-forming VOC upon reaction with OH (Supplementary Table 1). The VOC can be separated into three groups based on their HOM production capability (Fig. 3): (1) linear and branched alkanes (no observed HOM formation), (2) oxygenated VOC (high yields even at low oxidation rates), and (3) cycloalkanes (sharply increasing yield with increasing oxidation rate). For oxygenated VOC, the aldehyde functionality promotes the autoxidation better than an alcohol, as has been

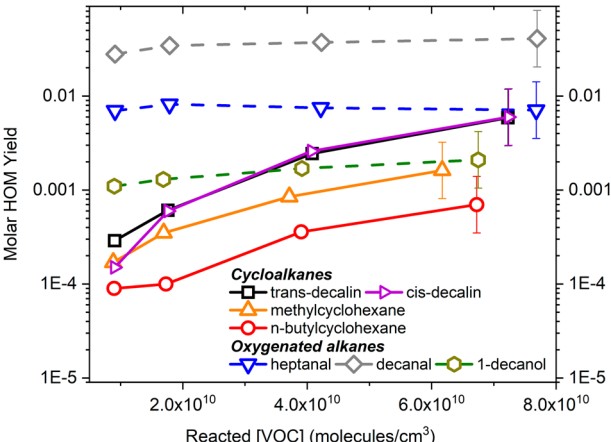

**Fig. 3 Molar HOM yields at 300 K as a function of reacted VOC, measured in the Helsinki flow reactor at a residence time of 3 s.** The error bars, included only for the last point of each trace, show the uncertainty of the HOM measurement. No multi-generation OH oxidation takes place due to the high VOC concentration (10 ppm) and the short residence time. Linear alkanes did not produce observable HOM signals under these conditions.

observed earlier[11,22], and an increased chain length further supports HOM formation. The HOM yield from decanal is a factor of 5 higher than that from heptanal, which corresponds with reported relative SOA yields between $n$-decane and $n$-heptane[23]. For cycloalkanes, the HOM yields increase in the order: decalin > methylcyclohexane > $n$-butylcyclohexane. This behavior qualitatively matches the findings that non-alkylated cyclic compounds tend to have higher SOA yields than the corresponding alkylcyclohexanes[17]. In total, our findings suggest that autoxidation and HOM formation may be critical for explaining the SOA yields for alkanes, just as has been shown earlier for many biogenic systems[4,24].

The strong increase in molar HOM yields from cycloalkanes with rising VOC conversion (Fig. 3) suggests that bimolecular $RO_2$ reactions are important for multi-step autoxidation. We utilized a simple kinetic model (Supplementary Note 7) in order to better understand the observed behavior depicted in Fig. 3. The model shows that the near-constant molar HOM yield from decanal as a function of reacted VOC can be explained (Supplementary Fig. 14) by assuming that the HOMs from decanal represent mainly autoxidation products that have undergone one RO isomerization step (in addition to $RO_2$ H-shifts). In contrast, the steeper increase in the case of decalin HOMs can be explained by requiring two RO steps (Supplementary Fig. 14). To an extent, the aldehydes (which are first-generation oxidation products of alkanes) "short circuit" one step of $RO_2 \rightarrow RO$ conversion by providing an oxygenated moiety that is primed for autoxidation but also vulnerable to fragmentation via C–C bond scission; this is consistent with the observed lower but non-zero SOA mass yields from aldehydes compared to alkane precursors of similar volatility[25]. Our findings on the importance of RO chemistry support earlier interpretations explaining the large differences in SOA yields observed from the different alkane groups[16]. However, with our new mechanistic insights, the differences can be directly linked to autoxidation and HOM formation, at considerably higher yields than earlier thought.

**Importance of bimolecular reactions.** For a deeper insight into alkane autoxidation, we conducted additional experiments in the

Leipzig flow reactor at TROPOS ("Methods"). Using a CI-APi-TOF with ethylaminium, $C_2H_5NH_3^+$, as the reagent ion, we were able to measure nearly all oxidized products, including the $RO_2$ radicals[19,26]. Results from the oxidation of decalin, with OH produced from the reaction of TME + $O_3$, show a striking sequence of $RO_2$ radicals bearing 2–9 O-atoms (Supplementary Fig. 15). Our kinetic model of the reaction system (Supplementary Note 7) with explicit representation (discounting isomers) of each $RO_2$ reproduces the observations by a combination of both RO and $RO_2$ isomerization steps.

The low precursor loadings and the limited residence time (7.9 s) in the Leipzig flow reactor do not allow the $RO_2$ reactions to run to completion, as they would in the atmosphere[19,26]. This means that many primary $RO_2$ radicals will simply exit the flow reactor, while a bimolecular reaction might have been able to initiate isomerization leading to a much more highly oxygenated species. To address this, we added NO at different concentrations to accelerate the $RO_2$ radical conversion via the $RO_2 + NO \rightarrow RO + NO_2$ reaction, which is a very typical $RO_2$ fate in urban environments. Here, the initiating OH radicals were produced via isopropyl nitrite photolysis, and we studied decalin, cyclohexane and $n$-decane oxidation. NO addition to the reaction system greatly influenced the concentration and distribution of oxidation products (Supplementary Note 8, Supplementary Figs. 16–18).

Figure 4a depicts how products with different levels of oxidation change as a function of the added NO. Contrary to the reported results on the autoxidation of many biogenic VOC[4], where NO suppresses autoxidation and HOM formation[4,27,28], the yields of many highly oxygenated products, especially in the case of decalin, increase with increasing NO concentrations, all the way up to $2.4 \times 10^{11}$ molecules cm$^{-3}$ (NO mixing ratio of about 10 ppb). This highlights the importance of the RO isomerization steps, and also indicates that $RO_2$ isomerization in these systems needs to be very fast to compete with termination reactions. Most strikingly, the molar HOM yield (i.e., products with 6 or more O-atoms) is much higher than expected, nearly 20%, being one of the highest yields reported for any VOC-oxidant system[10]. Overall, our findings regarding the influence of NO on the production of highly oxygenated species are in excellent agreement with reported SOA yields: while NO often decreases SOA yields from monoterpene ozonolysis, alkane SOA yields remain high at elevated NO. Further, for compounds with multiple rings (including some biogenic VOC like sesquiterpenes), the SOA yields even increase with NO[29–31]. This is yet another indication that autoxidation can be a major, heretofore unrecognized, driver of atmospheric SOA formation even for alkanes and even under highly polluted conditions.

For a more detailed look into the distribution of products from different alkane types at high NO, mass spectra are presented in Fig. 4b–d for decalin, cyclohexane and $n$-decane, respectively. At the highest NO concentration of $2.4 \times 10^{11}$ molecules cm$^{-3}$, a mixture of organic nitrates (blue), carbonyls (black) and $RO_2$ radicals (red) are visible. For decalin, we see products with up to 11 O-atoms, for cyclohexane up to 8, and even for $n$-decane, we see products with up to 6 O-atoms. For each of these precursors, we observed a much higher oxygen content in the products than previously thought[18], as was depicted already in Fig. 1. These results highlight the importance of multi-step isomerization of RO and/or $RO_2$ radicals for all types of alkanes, even under high NO concentrations.

We have shown that multi-step autoxidation of long-chain alkanes ($C_6$ and larger) is important under both combustion and atmospheric conditions. Under combustion conditions, we show that multi-step autoxidation driven by $RO_2$ radicals extends to a fourth and even a fifth $O_2$ addition, enriching our mechanistic understanding of fuel ignition chemistry. Under atmospheric

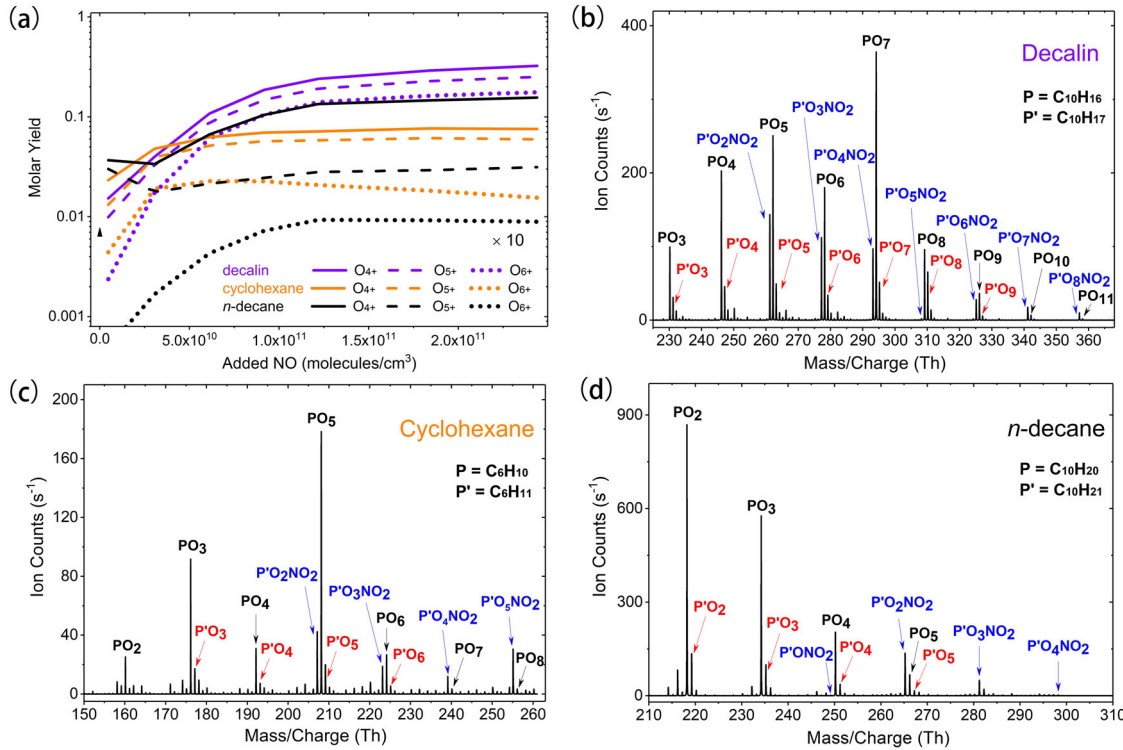

**Fig. 4 Effects of NO on product distributions from OH oxidation of decalin, cyclohexane and *n*-decane in the Leipzig flow reactor. a** The experimentally measured molar yields of products with at least 4, 5, or 6 O-atoms, respectively, are given as different line types, while the precursor alkanes each have separate line colors. The molar yield of products with at least 6 O-atoms from *n*-decane oxidation is multiplied by 10 for clarity. The data points were measured at NO concentration of $4.7 \times 10^9$, $3.1 \times 10^{10}$, $6.1 \times 10^{10}$, $9.2 \times 10^{10}$, $1.2 \times 10^{11}$, $1.8 \times 10^{11}$, and $2.4 \times 10^{11}$ molecules/cm$^3$, respectively. We note that this connection between NO and HOM yield should not be applied directly to the atmosphere, as the exact relations depend strongly on experimental conditions. The mass spectra, measured as adducts with ethylaminium, at the highest NO addition of $2.4 \times 10^{11}$ molecules cm$^{-3}$ are plotted separately for **b** decalin, **c** cyclohexane, and **d** *n*-decane. In each spectrum, radicals are in red, nitrates in blue, and other closed shell compounds, mainly carbonyls, in black.

conditions, we show that a single OH radical reaction can initiate autoxidation, driven by both RO and RO$_2$ isomerization reactions, for linear and, in particular, cycloalkanes. Multiple generations of OH oxidation take many hours in the atmosphere; they are not, as previously assumed, required to form highly oxygenated products able to contribute to atmospheric secondary organic aerosol. Instead, multi-step autoxidation can occur in seconds. However, we also show that aldehydes, as surrogates for first-generation oxidation products from alkanes, have high autoxidation potential, suggesting that also later generation OH oxidation will likely lead to very efficient autoxidation. Our results from different alkane structures, under varying NO$_x$ conditions, provide mechanistic explanations for the SOA yields observed in various systems, and emphasize the importance of RO isomerization reactions beside RO$_2$ isomerization steps. Earlier studies have observed considerable HOM yields from biogenic and aromatic precursors, often being able to link the autoxidation process directly to the SOA forming potential. This work now adds the only remaining major atmospheric SOA precursors, namely alkanes, as the last, and perhaps the most surprising, species class for which the importance of autoxidation has been identified.

## Methods
### Detection of oxidation products
*CI-APi-TOF.* For detection of oxygenated species and HOM with multiple peroxide functionalities in our experiments, we used the Chemical Ionization Atmospheric Pressure interface time-of-flight mass spectrometer (CI-APi-TOF[32]). The CI-APi-TOF consists of an Eisele-type[33] chemical ionization (CI) inlet coupled to an APi-

TOF mass spectrometer[34] (Tofwerk Ag/Aerodyne Research). The resolution of both instruments used in this study was ~4000.

In the Helsinki flow reactor experiments, nitrate ions (NO$_3^-$, selective toward HOM) were used as reagent ions, while in the Leipzig experiments protonated ethylamine (C$_2$H$_5$NH$_3^+$, selective to a broad range of oxidation products) was used[26,35,36]. In both ionization schemes, the oxygenated species are detected as clusters with the reagent ions.

### Experiments
*Jet stirred reactor, JSR.* The schematic of the jet stirred reactor (JSR) setup used for autoxidation investigations under combustion conditions is shown in Supplementary Fig. 1. The setup is similar to that presented in an earlier study[6]. An oven was used to control the temperature of the reactor. The air flow was delivered through the outer inlet tube, while the VOC was delivered by bubbling with N$_2$ and feeding the gas mixture through the inner inlet tube. The VOC and air are quickly mixed in a small volume prior to flowing into the four injector nozzles of the reactor. The mole fraction of *n*-decane, 2,7-dimethyloctane, *n*-butylcyclohexane, 2-decanone, and decanal was 110 ppm, 420 ppm, 200 ppm, 60 ppm, and 27 ppm, respectively. The rest of the flow is air. The reactor residence time was ~2.6 s. A quartz probe was adopted to sample the mixture at the reactor exit. The sample was diluted by 10 L air and analyzed by the CI-APi-TOF.

The JSR reactor was also used for the transition experiments from combustion to atmospheric conditions. In these experiments, TME and O$_3$ were used to produce OH radicals[37]. The TME and O$_3$ were diluted in air and fed into the reactor through the inner and outer inlet tubes, respectively. The mole fraction of decanal was 22 ppm and the OH concentration of $1.2 \times 10^7$ molecules/cm$^3$ at 300 K. The residence time was 3–5 s depending on the reactor temperature. A quartz probe was adopted to sample the mixture at the reactor exit. The sample was diluted by 10 L min$^{-1}$ (STP) air and analyzed by the CI-APi-TOF. A schematic of the system is shown in Supplementary Figure 4.

*Helsinki flow reactor.* A flow reactor at the University of Helsinki was used to study autoxidation under atmospheric conditions. The experiment was carried out in a flow reactor (1 meter long and 2.4 cm inner diameter) with a residence time of 3 s. The VOC concentration was ~10 ppm. OH radicals were generated by reacting

TME with $O_3$, and OH concentration was varied by varying the amount of TME in the reactor. The reaction products were analyzed by the $NO_3^-$ CI-APi-TOF. The signal of TME measured by the Vocus PTR-TOF-MS (Tofwerk Ag/Aerodyne Research)[38] was used to determine its consumption rate, and subsequently to estimate the OH concentration The experimental setup is shown in Supplementary Fig. 7. The method used to estimate HOM yields in the Helsinki flow reactor is described in detail in Supplementary Note 7.

*Leipzig free-jet flow reactor.* The experiments were performed in a free-jet flow system at T = 295 ± 2 K, the relative humidity is < 0.1%, and a pressure of 1 bar purified air that allowed investigations for nearly wall-free conditions[39]. The reaction time was 7.9 s. The free-jet flow system consists of an outer tube (length: 200 cm, inner diameter: 16 cm) and a moveable inner tube (outer diameter: 9.5 mm) with a nozzle of 3 mm inner diameter. Ozone premixed with air (5 L min$^{-1}$ STP) was injected through the inner tube into the main gas stream (95 L min$^{-1}$ STP), which contained the second reactant (TME) and the alkanes diluted in air. In the case of photolysis for OH radical generation, isopropyl nitrite was injected through the inner tube instead of ozone.

A proton transfer reaction - mass spectrometer (PTR-MS; HS PTR-QMS 500, Ionicon) served as an on-line monitor for organic reactants. Detection of peroxy radicals and closed-shell products was carried out by means of $C_2H_5NH_3^+$ CI-APi-TOF sampling from the center flow of the free-jet flow system with a rate of 10 L min$^{-1}$ (STP). Stated concentrations represent lower limit values obtained from a calculated calibration factor and include the duty cycle correction[19,26]. The sensitivity for compounds with only 3 or less O-atoms can be severely underestimated due to their weaker binding with the reagent ion ethylaminium.

The amount of reacted alkane from the photolysis experiments has been determined by measuring the $SO_3$ formation from the parallel reaction of OH radicals with $SO_2$[40]:

$$OH + alkane \rightarrow products \qquad (I)$$

$$OH + SO_2(+O_2) \rightarrow SO_3 + HO_2 \qquad (II)$$

## Data availability
All data are available in the main text or the Supplementary materials.

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

## Acknowledgements
This work was supported by the National Natural Science Foundation of China (Grant 51976208), National Key Research and Development Program of China (Grant 2019YFA0405602), King Abdullah University of Science and Technology Office of

Sponsored Research (Grant OSR-2016-CRG5-3022), the European Research Council (Grant 638703-COALA), the US National Science Foundation (Grant AGS1801897) and the Academy of Finland (Grants 299574, 326948, 307331, 317380, and 320094).

## Author contributions

Z.W., T.B., M.P.R., and P.R. conducted the measurements and analysis, L.Q. and O.G. provided data analysis tools, Z.W., M.E., S.M.S., and T.B. wrote the manuscript, M.E., Z.W., T.B., and O.G. provided model calculations. Z.W., M.E., M.P.R., S.M.S., and T.B. developed the mechanisms. N.M.D., L.X., and M.M-P. did additional data interpretation and editing of the manuscript. All authors discussed the results and commented on the paper.

## Competing interests

The authors declare no competing interests.
