## [Peer Review File · Communications Chemistry]

Reviewers' comments:

Reviewer #1 (Remarks to the Author):

The paper "From combustion engines to the atmosphere: unexpectedly efficient alkane oxidation" by Wang et al. investigates the oxidation processes of alkanes which lead to highly oxidized molecules HOMs, which are thought to be important for formation and growth of organic aerosol in the atmosphere, and compare the mechanistic systems in combustion engines and the atmosphere. They find in the pathway of autooxidation, which is known in combustion and recently also in atmospheric systems, for a range of nonlinear alkanes HOM production by autooxidation. The quantitative yields are determined with large uncertainty, but to the best state of current technology.

The experiments and analytic tools applied in the paper are state of the art and the obtained data seem to be very thoughtful and critical analyzed. This paper applies chemical models to understand detailed process steps in the investigated systems. A short come for transferring the results to atmospheric conditions is the very short reaction time (up to 3 sec) in the flow tube. Therefore, consecutive reactions are not observed. The details of the mechanistic investigation are partly new and original, especially the advanced analytic system, which enables to study the processes in more detail at much lower concentration and yields. The paper is interesting for the atmospheric chemistry community investigating in the fate of VOCs in the atmosphere. The claim of "unexpectedly efficient", through out the text and in the headline can be misleading, if you look at the determined yield of below 1%, or at least is it should be shown how large the contribution of this class of VOC to HOM formation in the atmosphere could be compared to other sources. It is not evident for me that this paper fits in this journal, though it is a very strong paper in quality with new results on the chemical mechanistic details of HOM formation for a group of VOCs the alkanes. The importance of RO radicals is in principal not new (Ziemann et al., 2009), but here it is shown by experimental evidence on a molecular level.

Below a few specific comments:

171-172 "For oxygenated alkanes, the aldehyde functionality promotes the autoxidation better than an alcohol, and an increased.." here reference (e.g. Mentel et al. 2015) has shown this in detail.

181-184 figure 3: some indication of the uncertainty of the estimated yields should be in the main text and figure. The description of the calibration and yield estimate in the supplement is clear and sufficient.

221 -224 " Contrary to the reported results on the autoxidation of biogenic VOC 4, where NO suppresses autoxidation and HOM formation, the yields of many highly oxygenated products, especially in the case of decalin, increase with increasing NO concentrations, all the way up to 2.4×10^{11} molecules cm^{-3} (NO mixing ratio of about 10 ppb)."

Pullinen et al. 2020 showed that high NO_x does not suppress autooxidation.

Pullinen, I., S. Schmitt, S. Kang, M. Sarrafzadeh, P. Schlag, S. Andres, E. Kleist, T. F. Mentel, F. Rohrer, M. Springer, R. Tillmann, J. Wildt, C. Wu, D. F. Zhao, A. Wahner, and A. Kiendler-Scharr (2020), Impact of NO_x on secondary organic aerosol (SOA) formation from alpha-pinene and beta-pinene photooxidation: the role of highly oxygenated organic nitrates, *Atmospheric Chemistry and Physics*, 20(17), 10125-10147, doi:10.5194/acp-20-10125-2020.

Mentel, T. F., M. Springer, M. Ehn, E. Kleist, I. Pullinen, T. Kurten, M. Rissanen, A. Wahner, and J. Wildt (2015), Formation of highly oxidized multifunctional compounds: autoxidation of peroxy radicals formed in the ozonolysis of alkenes - deduced from structure-product relationships, *Atmospheric Chemistry and Physics*, 15(12), 6745-6765, doi:10.5194/acp-15-6745-2015.

Reviewer #2 (Remarks to the Author):

The authors of this manuscript extend their previous research achievement of highly oxygenated intermediates from low-temperature combustion engine (2017, PANS) to the atmosphere, present the results about efficient oxidation of alkanes, including molar yields of highly oxygenated molecules (HOM), and discuss the formation mechanisms of autoxidation and HOM in detail. Although the alkane autoxidation in low-temperature combustion has been recorded in literature, the HOM formation from alkane in the atmosphere has not been reported. Thus, this manuscript represents a great progress in understanding alkane autoxidation and HOM formation, which is very important to air quality and climate change. Nevertheless, the authors have an attempt to link the combustion process to the atmosphere, which is probably not suitable. The low or intermediate-temperature combustion (<1000 K) is currently not available in the transport sector (vehicles). So far, no observation has been made about autoxidation products from vehicle exhaust, although the observed results of ambient HOM have been reported. Furthermore, the authors should probably remind readers that current results may not be applied directly to the atmosphere, since the experimental system is much far away from the real atmosphere. Indeed, decane produces HOM in the JSR experiment, but not in the Helsinki experiment, from which it is known that experimental conditions (or reactors) may greatly affect the reaction processes. In fact, the authors have not mentioned the experimental condition of humidity that is very important to the reaction processes and HOM formation, except for the experiments in the Leipzig flow reactor (Figure S16-S18, <0.1% RH). If possible, the authors should discuss the influences of RH on the alkane autoxidation.

Specific comments:

Fig. 2. There is an obvious difference in signal intensity of products from decanal between 520 and 392 K. These signals do not increase with increasing temperature in the left panels. It seems there is a change in a specific temperature. Any explanation?

Fig.3 (Fig. S13). Linear alkanes do not produce observable HOM signals. Is this due to experimental conditions such as temperature and VOC concentrations?

In Supplementary Information, Scheme 2 has a typing error (+HO₂).

In Schemes S2 and S3, why is the rate constant of H-shift+ O₂ different, which is larger than the rate constant of scission+O₂? If these constants are not based on O₂ concentration and the value of the rate constant for the reaction of O₂ with alkoxy radicals, how are these estimated?

Page 27. There is probably a typing error in equation $dC_3H_5O_3/dt$.

Reviewer #3 (Remarks to the Author):

This manuscript reports three sets of experiments on the rapid oxidation (over 3 – 8 s) of C₆-C₁₀ alkanes and carbonyl compounds from 550 K to 300K, in which low-vapor-pressure, highly oxygenated products (HOMs) were observed. The obtained HOM yields led the authors to suggest that autoxidation reactions should be fast for alkanes in the atmosphere, unlike what was previously thought, and a potential major, yet unrecognized, contributor to Secondary Organic Aerosols (SOA) formation from these compounds in the atmosphere, even under polluted conditions. In particular, the HOM yield was found to increase from cyclic to non-cyclic alkanes and with added NO (thus polluted conditions).

This work is interesting and experimentally sound but its originality and importance for atmospheric chemistry are somewhat overstated: previous experimental and theoretical works on the autoxidation of alkanes, mitigating the originality of the present work, have been overlooked and need to be discussed; the large alkanes studied in this work are fairly rare in the atmosphere,

thus limiting their relevance for atmospheric chemistry; the conclusions drawn in this work on the autoxidation of alkanes in the atmosphere are too general as they apply only to the large compounds studied, while previous works concluded differently for smaller alkanes; finally the potential contribution of HOMs to SOA formation from alkanes, while an interesting hypothesis, is not demonstrated in this work as SOA were not monitored, and would have to be established in future works.

The present manuscript is recommended for publication once the comments (in the attached document) have been taken into account and the corresponding changes made in the text.

In accordance with comments from reviewers (*in italics below*), considerable revision has been made to this manuscript (COMMSCHEM-20-0333-T). Responses appear below in blue; corresponding changes in the manuscript are highlighted in green.

For Reviewer 1:

We appreciate the positive comments from this reviewer. We have responded to their suggestions for improvement below.

1. The experiments and analytic tools applied in the paper are state of the art and the obtained data seem to be very thoughtful and critical analyzed. This paper applies chemical models to understand detailed process steps in the investigated systems. A short come for transferring the results to atmospheric conditions is the very short reaction time (up to 3 sec) in the flow tube. Therefore, consecutive reactions are not observed.

Reply: We respectively disagree with the assertion that a short experimental timescale precludes “atmospheric relevance”. There is a long tradition of studying elementary chemical reactions on the μs -ns timescale for direct application to atmospheric chemical mechanisms; most of the rate coefficients used in atmospheric mechanisms come from such experiments. Autoxidation quickly sets off a number of sequential and parallel reactions that makes elementary details difficult or nearly impossible to fully unravel. The chosen residence time enables to follow the autoxidation reactions. We thus access “one generation” of autoxidation, and in this way can map out the initial steps forming organic radicals that will eventually form stable reaction products. We are not claiming in any way that we map out all of the oxidation, or even autoxidation, under atmospheric conditions. However, consecutive reactions would likely only serve to increase the oxidation levels even further, and thus the surprisingly fast oxygenation that we report is, in fact, more of a lower limit. We will clarify in the paper that our work focuses on the initial steps of autoxidation, **line 179-180** in the revised paper. In addition we would like to point out that atmospheric chemists solved the entirety of gas-phase stratospheric chemistry, and really all of it, with kinetics experiments having either 100 ms (flowtube) or 100 ns (flash photolysis) timescales.

2. The claim of “unexpectedly efficient”, through out the text and in the headline can be misleading, if you look at the determined yield of below 1%, or at least is it should be shown how large the contribution of this class of VOC to HOM formation in the atmosphere could be compared to other sources.

Reply: Conventional knowledge suggests that atmospheric autoxidation requires suitable structural features, like double bonds or oxygen-containing moieties, in the precursors (Bianchi et al., 2019). With neither of these

functionalities, alkanes are thought to have minor susceptibility to extensive autoxidation. Isomerization of RO radicals has been shown (Ziemann et al., 2009), but this pathway terminates quickly, in the case of decane yielding a keto-alcohol with two O-atoms. This work shows that alkanes do undergo autoxidation, which is much more efficient than previously thought. In the case of decane, we identify around 15% of the products to include 4 or more O-atoms (Fig. 4a in the text). For decalin, the HOM yield (i.e. products with 6 or more O-atoms) at high loadings of NO is nearly 20%, being one of the highest yields reported for any VOC-oxidant system. These are the “unexpected” features of our work, which we try to highlight. We have now tried to make these features more clear in our revised manuscript (**line 180-182, and line 244-245**).

2. It is not evident for me that this paper fits in this journal, though it is a very strong paper in quality with new results on the chemical mechanistic details of HOM formation for a group of VOCs the alkanes. The importance of RO radicals is in principal not new (Ziemann et al., 2009), but here it is shown by experimental evidence on a molecular level.

Reply: Thanks for the positive comments of this work. We agree with the reviewer that the involvement of RO radicals was mentioned in the work of Ziemann et al., as acknowledged in our previous response above. However, these early studies have in no way realized the full potential of sequential RO isomerization. In their work, the isomerization products from RO radicals were quickly terminated, and thus the SOA formation pathways were assumed to be from multigeneration OH oxidation reactions. In this work, we reveal the importance of RO driven autoxidation in HOM formation, forming condensable products at high yields already from one OH reaction, i.e., “one generation” of oxidation. Based on these critical conceptual advances in the reaction mechanism, compared to e.g. the work of Ziemann et al., 2009, we believe our work fits well in this journal.

We already motivate in our manuscript (**lines 86-98**) that earlier RO chemistry was likely misinterpreted, but we have now clarified this to a mechanism driven by RO autoxidation.

3. a few specific comments

(a) 171-172 “For oxygenated alkanes, the aldehyde functionality promotes the autoxidation better than an alcohol, and an increased..” here reference (e.g. Mentel et al. 2015) has shown this in detail.

Reply: Thank you, we have now added references to the relevant section (**line 190**):

“For oxygenated alkanes, the aldehyde functionality promotes the autoxidation better than an alcohol, as reported earlier (Mentel2015, bianchi2019), and...”

(b) 181-184 figure 3: some indication of the uncertainty of the estimated yields should be in the main text and figure. The description of the calibration and yield estimate in the supplement is clear and sufficient.

Reply: Error bars have been added in the revision, to show the uncertainty.

(c) 221 -224 “ Contrary to the reported results on the autoxidation of biogenic VOC 4, where NO suppresses autoxidation and HOM formation, the yields of many highly oxygenated products, especially in the case of decalin, increase with increasing NO concentrations, all the way up to 2.4×10^{11} molecules cm^{-3} (NO mixing ratio of about 10 ppb).”

Pullinen et al. 2020 showed that high NO_x does not suppress autoxidation.

Reply: We are happy to learn of this new reference. We feel the statement that “high NO_x does not suppress autoxidation” is generalizing the results of Pullinen et al. too far. First, the paper does show a decrease, although only by about 1/3, for HOM formation from the α -pinene photo-oxidation system when NO_x is added, compared to NO_x-free conditions. Thus, we believe that our initial statement is still valid, that reported results show a decrease in autoxidation. Second, photo-oxidation of the two specific compounds studied by Pullinen et al (alpha- and beta-pinene) will both primarily form RO₂ with the peroxy group on a ring, which we here show to be a good system for further autoxidation, if NO reactions convert the RO₂ into an RO, which can break the ring. However, this comment by the reviewer does highlight the limitation of grouping molecules into “biogenics” and “anthropogenics”, as from the point-of-view of chemical reactions, there can be many similarities between molecules in the two groups, as well as large differences between molecules within the same group. As an example, which is also noted briefly in our manuscript, some biogenic sesquiterpenes are in practice polycyclic alkanes, and these have been shown to have SOA yields that increase with NO. We have therefore changed the wording in several places in the manuscript to avoid overly generalizing this separation between VOC of biogenic and anthropogenic origin. We have also included more references, including the one mentioned by the reviewer.

“Contrary to the reported results on the autoxidation of many biogenic VOC⁴, where NO often suppresses autoxidation and HOM formation (Ehn2014, Pullinen2020, Roldin2019 NatComm), the...” (Line 239-240)

For Reviewer 2:

We appreciate the positive comments from this reviewer. We have responded to their suggestions for improvement below.

1. This manuscript represents a great progress in understanding alkane autoxidation and HOM formation, which is very important to air quality and climate change. Nevertheless, the authors have an attempt to link the combustion process to the atmosphere, which is probably not suitable. The low or intermediate-temperature combustion (<1000 K) is currently not available in the transport sector (vehicles). So far, no observation has been made about autoxidation products from vehicle exhaust, although the observed results of ambient HOM have been reported.

Reply: We are not sure what the reviewer exactly means by this. The reviewer seems to suggest that engines do not see temperatures below 1000 K, which is not true. There are a plethora of papers discussing low temperature combustion conditions (well below 1000 K, and down to 500 K) encountered in engines in phenomena related to knock and pre-ignition in SI engines [Prog Energy Combust Sci. 2011, 37, 741-83; Prog Energy Combust Sci. 2009, 35, 398-437; Prog Energy Combust Sci. 2015, 46, 12-71]. Modern advanced combustion engines (e.g., Mazda SkyActiv X) easily see much lower temperatures. In addition, the fact that HOMs are never measured in the exhaust is irrelevant. We are looking at what happens inside the engine during the initial steps of the ignition/combustion process, and not in the engine exhaust. Furthermore, as we show a ramp from 530K to 300K in Fig.2, any oxidation that might happen there would then follow pathways that we report in Fig. 2. Our main intent in addressing the autoxidation process over a range of conditions ranging from combustion to atmospheric oxidation is not so much to relate the emissions from engines to the subsequent atmospheric fate of those emissions as to connect the fundamental chemical understanding across the temperature range (and to connect the disciplines of combustion chemistry and atmospheric chemistry). That is why we feel that a journal focused on fundamental chemistry, such as Communications Chemistry, is appropriate for our manuscript. We would also like to note that so far, no measurements at all have been reported from vehicle exhaust using this type of instrumentation, and therefore the lack of HOM observations cannot be seen as an indication that this chemistry would not be relevant.

2. The authors should probably remind readers that current results may not be applied directly to the atmosphere, since the experimental system is much far away from the real atmosphere.

Reply: We agree with the reviewer. We have added this reminder in the revised paper. **Lines 259-260.**

3. the authors have not mentioned the experimental condition of humidity that is very important to the reaction processes and HOM formation, except for the experiments in the Leipzig flow reactor (Figure S16-S18, <0.1% RH). If possible, the authors should discuss the influences of RH on the alkane autoxidation.

Reply: We appreciate the research idea. The humidity of the Helsinki experiment is estimated to be <1% RH

(line 177). However, based on our experience and chemical intuition, we have no reason to expect that RH would have a large effect on autoxidation, and this has also been shown recently by Li et al. (Atmos. Chem. Phys., 19, 1555–1570, 2019).

4. a few specific comments

(a) Fig. 2. There is an obvious difference in signal intensity of products from decanal between 520 and 392 K. These signals do not increase with increasing temperature in the left panels. It seems there is a change in a specific temperature. Any explanation?

Reply: Fig 2a shows the decanal autoxidation at combustion relevant conditions. By heating the reactor to 520 K, the oxidation and OH radical generation is initiated through decanal + O₂ reactions. In Figures 2b and 2c, at temperatures 392 and 334 K, decanal can no longer react with O₂, and thus we added TME and O₃ into the reactor to produce OH radicals. Therefore, the exact oxidation rates are unknown and Figures 2a and 2b-c should not be compared quantitatively. We have added a sentence in the revised text to make this clear. **Line 175-176.**

(b) Fig.3 (Fig. S13). Linear alkanes do not produce observable HOM signals. Is this due to experimental conditions such as temperature and VOC concentrations?

Reply: The reviewer is correct, temperature, VOC and OH concentrations, and residence time all affect the autoxidation process. If we increase the OH concentration and residence time to allow for multigeneration OH oxidation, we expect to observe HOM signals also from linear alkanes. However, at room temperature, a single OH reaction does not seem to be able to produce noticeable HOM (i.e. molecules with 6 or more O-atoms) concentrations, as shown in Figures 4a and 4d.

(c) In Supplementary Information, Scheme 2 has a typing error (+HO₂).

In Schemes S2 and S3, why is the rate constant of H-shift+ O₂ different, which is larger than the rate constant of scission+O₂? If these constants are not based on O₂ concentration and the value of the rate constant for the reaction of O₂ with alkoxy radicals, how are these estimated?

Page 27. There is probably a typing error in equation $dC_3H_5O_3/dt$.

Reply: The rate constant of H-shift + O₂ and scission+O₂ are taken directly from the MCM model. There are two steps in the above reactions, i.e., H-shift and O₂ addition, scission and O₂ addition. The O₂ addition to the radical site is much faster than the H-shift and scission reactions. Thus, only the rate constant for H-shift and scission is shown in the figure.

The typos have been corrected.

For Reviewer 3:

We appreciate the positive comments from this reviewer. We have responded to their suggestions for improvement below.

1. This work is interesting and experimentally sound but its originality and importance for atmospheric chemistry are somewhat overstated: previous experimental and theoretical works on the autoxidation of alkanes, mitigating the originality of the present work, have been overlooked and need to be discussed; the large alkanes studied in this work are fairly rare in the atmosphere, thus limiting their relevance for atmospheric chemistry; the conclusions drawn in this work on the autoxidation of alkanes in the atmosphere are too general as they apply only to the large compounds studied, while previous works concluded differently for smaller alkanes.

Reply: We appreciate these comments. Below, we have addressed them one by one.

(1a) Previous experimental and theoretical works

Several works on the autoxidation of RO₂ and RO from alkanes have been overlooked and need to be included in the discussion. In addition, it is surprising that the kinetic modeling in this work is based on MCM, which does not include autoxidation reactions or other recent update on RO₂ reactions. A more relevant model for RO₂ reactions is suggested below.

Experimental investigation of the first steps of the autoxidation C4 – C8 alkanes:

Noziere and Vereecken, Direct Observation of Aliphatic Peroxy Radical Autoxidation and Water Effects:

An Experimental and Theoretical Study *Angew. Chem. Int. Ed.*, 58, 13976 – 13982; doi:

10.1002/anie.201907981, 2019.

SAR for the autoxidation rate constants of a wide range of RO₂:

Vereecken and Nozière, H migration in peroxy radicals under atmospheric conditions, *Atmos. Chem.*

Phys., 20, 7429; doi:10.5194/acp-20-7429-2020; 2020.

SARs for the autoxidation and decomposition of alkoxy radicals:

Vereecken, L. and Peeters, J.: Decomposition of substituted alkoxy radicals—part I: a generalized

structure–activity relationship for reaction barrier heights, *Phys. Chem. Chem. Phys.*, 11(40), 9062–9074, doi:10.1039/b909712k, 2009.

Vereecken, L. and Peeters, J.: A structure–activity relationship for the rate coefficient of H-migration in substituted alkoxy radicals, *Phys. Chem. Chem. Phys.*, 12(39), 12608–12620, doi:10.1039/c0cp00387e, 2010.

For the kinetic modelling of RO₂ reactions:

Jenkin et al., Estimation of rate coefficients and branching ratios for reactions of organic peroxy radicals for use in automated mechanism construction, *Atmos. Chem. Phys.*, 19, 7691–7717; doi:10.5194/acp-19-7691-2019; 2019.

Reply: We have now included references to these papers in our revised manuscript. **Line 180-182** in the main text, and lines 39-41 and line 73 in the SI. **From this set of experiments, despite the short reaction time and low temperature, high signals of highly oxygenated species are still observed. This is unexpected, as RO₂ H-shifts in alkanes are reportedly slow and should rapidly lead to radical termination^{20,21}**

The purpose of the kinetic modeling in our work is simply to show that our results make sense with reasonable rate assumptions. There was no reasonable way to make an accurate full kinetic model since the reaction mechanisms of alkane autoxidation to form HOM remains largely unknown. We hope and expect that our findings will spur further work in this area, and someone venturing to make an explicit model of the type that the reviewer suggest. For our work, such an effort would clearly be out of scope.

The reviewer incorrectly implies that our kinetic modeling would have been based on MCM. The only usage of the MCM is to show typically used reactions schemes, e.g. in Section S1 and S2, where we show a general picture of the peroxy radical and alkoxy radical chemistry.

In Section S7, we present the kinetic modelling of decanal and decalin oxidation in flow reactors. (1) “Lumped” model for Helsinki experiments. In this model, as we have stated in the manuscript, there were no autoxidation reactions, and the only rate coefficient this model took from any external source was the RO₂ + RO₂ rate coefficient, which was “loosely based on the recent findings of Berndt et al (*Angew. Chem. Int. Ed.* 57, 3820-3824)”. (2) “Explicit” model for Leipzig experiments for decalin oxidation. In this model, as we also have stated in the manuscript, we just include the reactions of the RO₂ radicals; we could not distinguish the

structures of those RO₂ radicals. These reactions are not elemental reactions, thus we could not assign their rate coefficients from a specific reaction in the literature. The values are estimated to fit our experimental observations.

(1b) Relevance of the large alkanes studied to atmospheric chemistry

As mentioned above, the most abundant alkanes in the atmosphere are the smallest ones (C₁-C₆). Those studied in the present work, although potentially relevant for combustion systems, are less representative of atmospheric chemistry. While such compounds might indeed lead to fast peroxy and alkoxy autoxidation reactions, this is not the case for the smaller alkanes. Therefore, concluding on the importance of autoxidation for all alkanes in the atmosphere is a bit too general. And the importance of the present work for atmospheric chemistry is a bit overstated.

Reply: We agree with the reviewer's comments that the most abundant alkanes in the atmosphere are the smallest ones (C₁-C₆). While these most likely will be the most important for atmospheric chemistry in the sense of, for example, OH reactivity or ozone formation, they have negligible SOA yields (Lim & Ziemann, 2009). However, the long-chain alkanes, such as C₁₀ alkanes are also important emission from the evaporation of gasoline and diesel fuels, and their SOA yields have been reported to reach as high as 65%, depending on the structure (Lim & Ziemann, 2009). In fact, in a very recent paper, Wang et al. found that alkanes larger than C₁₀ are the largest source of SOA in the Pearl River Delta in China (see figure below), with the contribution from smaller alkanes being negligible.

Figure 9a from Wang et al. (2020), <https://doi.org/10.5194/acp-20-14123-2020>.

As such, we do not feel that we overstate the atmospheric importance, as we highlight the atmospheric impacts

of our work in the context of SOA and urban air quality, which is largely determined by particulate matter (PM). To be more clear, we have added to both the abstract and the conclusion paragraph that our results apply to long-chain alkanes (**line 43, and line 273**). In addition, the long-chain aldehydes are also important emission from the cooking of seed oils. Please refer to the series work of Schauer et al. (Environment Science and Technology, 1999, 33, 1578; 2002, 36, 1169; 2002, 36, 567).

Our original manuscript did not claim that our results were crucial for all atmospheric chemistry; we have now clarified better that our findings relate to C6-C10 alkanes. We also had noted on **line 191** that “an increased chain length further supports HOM formation”.

(1c) smaller comments

a) It would be useful to present Table S1, summarizing all the experiments, in the main manuscript, as the discussion of the three different sets of experiments, performed with different precursors, can be a little difficult to follow.

Reply: Thank you, this is a great suggestion, and we have now added it to the main text

b) In the legend of Fig. 1, both occurrences of ref. 18 seem to be a mistake, as ref. 18 is the MCM model.

Reply: The reference is right, but the first reference was badly placed and could be misinterpreted to think that it concerned the entire figure. We have removed the first instance.

c) In Fig. 4a: are the yield curves resulting from experimental measurements or modeling? Panels b, c, and d suggest that these are experimental results but, in that case, it would be important to indicate clearly the experimental points.

Reply: The curves are indeed from the experiments. We have made it clear now in the figure caption. **Line 257-260.** “The **experimentally measured** molar yields of products with at least 4, 5, or 6 O-atoms, respectively, are given as different line types, while the precursor alkanes each have separate line colors. The molar yield of products with at least 6 O-atoms from *n*-decane oxidation is multiplied by 10 for clarity. **The data points were measured at NO concentration of 4.7×10^9 , 3.1×10^{10} , 6.1×10^{10} , 9.2×10^{10} , 1.2×10^{11} , 1.8×10^{11} , and 2.4×10^{11} molecules/cm³, respectively.”**

We did not change the lines to markers since we think the use of different lines is the clearest way to show the trends of products with 4, 5, or 6 O-atoms for different VOC systems.

2. *finally the potential contribution of HOMs to SOA formation from alkanes, while an interesting hypothesis, is not demonstrated in this work as SOA were not monitored, and would have to be established in future works.*

Reply: This direct comparison is indeed an important one, and we are planning experiments to address this. For this paper, we *propose the link* between SOA and HOM as a hypothesis, as we see a very good correlation between our observed HOM yields (both as a function of alkane structure and NO concentration) with earlier reports on SOA yields.

We appreciate the reviewers' careful inspection of this manuscript and their insightful comments. All suggestions and comments have been seriously considered in the revision process, especially as regards prior knowledge and the relevance to the atmosphere. We feel that the manuscript has been significantly improved by these revisions and by clarifying all issues raised by the reviewers and the editor, and we hope that it is now suitable for publication in Communications Chemistry.

Yours Sincerely,

Zhandong Wang and Mikael Ehn

REVIEWERS' COMMENTS:

Reviewer #1 (Remarks to the Author):

I am satisfied with the rebuttal and correction / additions to the article.
No further comments, publish as is

Reviewer #2 (Remarks to the Author):

The authors of the manuscript have made the response to my comments, and modified the relevant contents. The authors have also modified the MS based on the comments by other reviewers. These efforts generally improve the quality of the manuscript. The paper will be greatly helpful to atmospheric chemistry community for further understanding of the role of alkanes in the formation of SOA. Thus, I am inclined to advise acceptance of this manuscript.

Reviewer #3 (Remarks to the Author):

I thank the authors for their answers and corresponding changes in the manuscript, which have improved the clarity of the presented work. To summarize their main findings, cyclic alkanes produce significant yields of highly-oxidized products under all NO_x conditions while for linear alkanes these yields strongly depend on the NO_x conditions: under Low-NO_x conditions the yields remain small until the formation of the first aldehyde products from RO₂+RO₂ chemistry, which is consistent with previous works on the first autoxidation steps of linear alkanes. By contrast, these yields are large under high-NO_x conditions, where RO chemistry is favored.

I do not have any further questions on the scientific content of the work. But I do have one last (but important) request concerning the nomenclature of some compounds, which are in contradiction with a basic IUPAC definition (and I apologize for missing it in the previous version of the manuscript): p. 8, li. 188 and 189 refer to "oxygenated alkanes" instead of "aldehydes". IUPAC is very clear, there is no such thing as "oxygenated alkanes" since alkanes are defined as compounds composed exclusively of C and H atoms (<https://doi.org/10.1351/goldbook.A00222>). Sorry to insist on this, but referring to aldehydes as a sub-class of alkanes would make any organic chemist (and most other chemists) jump. Please, replace "oxygenated alkanes" by "aldehydes". Anyway, these terms are confusing as they sound as if aldehydes were studied here as a class of alkanes, while it is clear in the remainder of the study that they were studied as oxidation products from alkanes.

Finally, I have a minor comment concerning the designation of the present work as "inter-disciplinary" in the abstract (p.2, Li. 44/45). I understand that the authors refer to combining knowledge from combustion chemistry and atmospheric chemistry. Still, the tools are similar and the ground discipline is chemistry in both cases. I am wondering if "trans-disciplinary" would not be more appropriate here.

Apart from these comments, I do not have any objection against publishing this manuscript.

We would like to thank the three reviewer's effort to improve the manuscript. We hope that it is now suitable for publication in Communications Chemistry after considering the comments of the last reviewer.

For Reviewer 3:

1. I do not have any further questions on the scientific content of the work. But I do have one last (but important) request concerning the nomenclature of some compounds, which are in contradiction with a basic IUPAC definition (and I apologize for missing it in the previous version of the manuscript): p. 8, li. 188 and 189 refer to “oxygenated alkanes” instead of “aldehydes”. IUPAC is very clear, there is no such thing as “oxygenated alkanes” since alkanes are defined as compounds composed exclusively of C and H atoms (<https://doi.org/10.1351/goldbook.A00222>). Sorry to insist on this, but referring to aldehydes as a sub-class of alkanes would make any organic chemist (and most other chemists) jump. Please, replace “oxygenated alkanes” by “aldehydes”. Anyway, these terms are confusing as they sound as if aldehydes were studied here as a class of alkanes, while it is clear in the remainder of the study that they were studied as oxidation products from alkanes.

Reply: Thanks for the valuable comment. We have changed oxygenated alkanes to oxygenated VOCs in the revised manuscript.

2. I have a minor comment concerning the designation of the present work as “inter-disciplinary” in the abstract (p.2, Li. 44/45). I understand that the authors refer to combining knowledge from combustion chemistry and atmospheric chemistry. Still, the tools are similar and the ground discipline is chemistry in both cases. I am wondering if “trans-disciplinary” would not be more appropriate here.

Reply: Thanks for this comment. We had a look of the meaning of inter-disciplinary to trans-disciplinary. It looks like they do not have much difference. We prefer to use inter-disciplinary.

Yours Sincerely,

Zhandong Wang and Mikael Ehn